# CARMIL: Context-Aware Regularization on Multiple Instance Learning models for Whole Slide Images

**Thiziri Nait Saada**[*†‡]                                                   NAITSAADAT@MATHS.OX.AC.UK
**Valentina Di-Proietto** [*†]                                      VALENTINA.DI-PROIETTO@OWKIN.COM
**Benoit Schmauch**[†]                                                  BENOIT.SCHMAUCH@OWKIN.COM
**Katharina Von Loga**[†]                                           KATHARINA.VONLOGA@OWKIN.COM
**Lucas Fidon**[†]                                                          LUCAS.FIDON@OWKIN.COM

**Editor:**

## Abstract

Multiple Instance Learning (MIL) models have proven effective for cancer prognosis from Whole Slide Images. However, the original MIL formulation incorrectly assumes the patches of the same image to be independent, leading to a loss of spatial context as information flows through the network. Incorporating contextual knowledge into predictions is particularly important given the inclination for cancerous cells to form clusters and the presence of spatial indicators for tumors. State-of-the-art methods often use attention mechanisms eventually combined with graphs to capture spatial knowledge. In this paper, we take a novel and transversal approach, addressing this issue through the lens of regularization. We propose Context-Aware Regularization for Multiple Instance Learning (CARMIL), a versatile regularization scheme designed to seamlessly integrate spatial knowledge into any MIL model. Additionally, we present a new and generic metric to quantify the *Context-Awareness* of any MIL model when applied to Whole Slide Images, resolving a previously unexplored gap in the field. The efficacy of our framework is evaluated for two survival analysis tasks on glioblastoma (TCGA GBM) and colon cancer data (TCGA COAD).

## 1 Introduction

The digitization of histopathology Whole Slide Images (WSIs) and the development of deep learning methods has lead to promising computational methods for cancer prognosis. One computational challenge is the large size of WSIs, of the order of $100,000 \times 100,000$ pixels. Processing images of such size with a deep neural network directly is not possible with the GPUs commonly available. Overcoming this problem, previous work proposes to tessellate each WSI into thousands of smaller images called tiles and global survival prediction per slide is obtained in two steps. The tiles are first embedded into a space of lower dimension using a pre-trained feature extractor model, and a MIL model is trained to predict survival from the set of tiles embeddings of a WSI (Herrera et al., 2016).

One limitation of MIL is the assumption that tiles from the same WSI are independent (Ilse et al., 2018). In particular, MIL models fail to leverage spatial interactions between tiles and their ordering in a WSI. In contrast, pathologists take into account the spatial

---

*. These authors contributed equally to this work
†. Owkin, Inc., New York, NY, USA
‡. Mathematical Institute, University of Oxford, Oxford, UK

organization of WSIs in their analysis. To tackle this issue, a correlated variant of MIL has been proposed in Shao et al. (2021). In practice, interactions between neighboring tiles in a WSI can be modeled using a graph of tiles. Graph Neural Network (GNN) (Kipf and Welling, 2016b; Meng and Zou, 2023) and Transformers with local attention (Reisenbüchler et al., 2022; Fourkioti et al., 2023) were proposed to capture correlations between neighboring tiles. However, as these correlations are parameterized using more complex architectures involving a greater number of parameters, the effectiveness of such mechanisms might be limited by the amount of available training samples. Besides, histopathology datasets usually contain only up to a few hundred WSIs. By contrast, incorporating spatial correlations between tiles into the model through regularization has been under-explored in computational pathology.

In this paper, we introduce the CARMIL framework to enhance the *Context-Awareness* of any MIL model, initially agnostic to the spatial context, using *Regularization*. In CARMIL, a spatial encoder and a spatial decoder are added between the feature extractor and the MIL model. In addition, a Context-Aware Regularization (CAR) loss function is proposed to train the spatial decoder to reconstruct the input graph of tiles. The proposed spatial encoder aims at distilling the spatial relation between neighboring tiles directly into the tile embeddings so that any MIL model can exploit this information. To this end, the spatial decoder and the CAR loss are designed to encourage bringing closer the embeddings learnt by the spatial encoder for tiles that are spatially close in the original WSI. To the best of our knowledge, such regularization strategy has so far not been studied in computational pathology. The nearest related work is the topologically-aware regularization for cancer blood cell classification introduced by Kazeminia et al. (2023). Yet, this regularization is based on the $0^{\text{th}}$ order Betti number, which only captures the connected components of the graph of tiles. Instead, CARMIL aims to reconstruct the entire graph structure.

Our main contributions are three-fold. First, we introduce CARMIL, a Context-Aware Regularization module based on Graph AutoEncoders that we incorporate into the classical MIL framework through regularization. Second, we propose DeltaCon, a novel metric to quantitatively assess the *Context-Awareness* of any tiles embedding. This metric allows for a better inspection of how well the embeddings conform with the original spatial arrangement of the tiles within the WSIs. Lastly, we evaluate CAR on a set of benchmark MIL models for the task of overall survival prediction for glioblastoma and colon cancer. These models are precisely chosen because they are originally context-independent, and we show through extensive quantitative and qualitative ablation studies how CARMIL turns them into Context-Aware models and leads to improved C-index.

## 2 Methods

### 2.1 Background: classical MIL framework for risk prediction

In this section, we summarize the main steps of the classical MIL pipeline, that we enhance with our Context-Aware Regularization in the next section.

**Preprocessing.** Each WSI is segmented to keep only the tissue and remove the background (e.g. using the otsu method). The tissue parts are then tessellated into tiles of size $224 \times 224$ pixels taken at 20X magnitude ($0.5\mu m$/pixel) and each WSI is reduced to a set

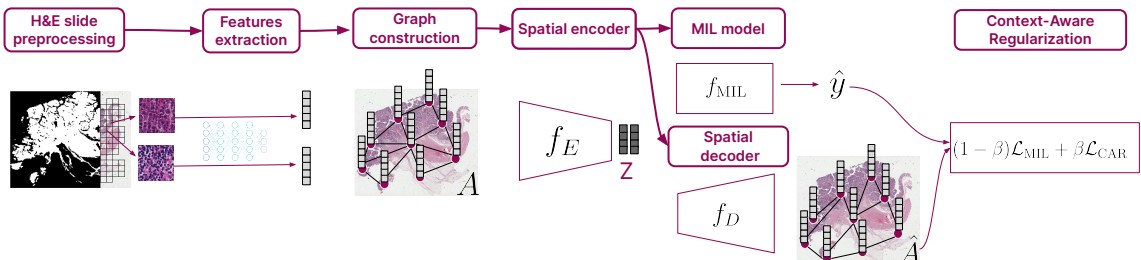

Figure 1: Our proposed framework, CARMIL, enhances any MIL model by incorporating spatial information through Context-Aware Regularization. In CARMIL, tiles embeddings are finetuned to allow for graph reconstruction by a spatial decoder.

of $n$ tiles of tissue selected at random. A pre-trained feature extractor is then used to map each tile into a low-dimensional tile feature space of dimension $d$. This results in a matrix representation $X^{(i)} \in \mathbb{R}^{n \times d}$ for the WSI indexed by $i$.

**MIL model.** The MIL model $f_{\mathrm{MIL}}$ is a parametric model trained to aggregate the tile features of a WSI and estimate a risk associated with the overall survival of the patient

$$f_{\mathrm{MIL}} : \mathbb{R}^{n \times d} \to \mathbb{R}. \tag{1}$$

The parameters $\theta_{\mathrm{MIL}}$ of the MIL model $f_{\mathrm{MIL}}$ are optimized to minimize a survival loss $\mathcal{L}_{\mathrm{MIL}}$, such as the Cox loss. Formally, this corresponds to the optimization problem

$$\min_{\theta_{\mathrm{MIL}}} \frac{1}{N} \sum_{i=1}^{N} \mathcal{L}_{\mathrm{MIL}} \left( f_{\mathrm{MIL}}(X^{(i)}; \theta_{\mathrm{MIL}}), y^{(i)} \right), \tag{2}$$

where $N$ is the number of training WSIs and $y^{(i)}$ is the overall survival risk of patient $i$. Examples of such MIL models are ABMIL (Ilse et al., 2018), Chowder (Courtiol et al., 2018), and AdditiveMIL (Javed et al., 2022). It is worth noting that the spatial relationships between tiles are not used by $f_{\mathrm{MIL}}$, since their ordering is random in $X^{(i)}$.

## 2.2 Context-Aware Regularization (CAR)

We propose a general approach to leverage local spatial context in computational histopathology. Our pipeline, which can be applied to any task or MIL model, is depicted in fig. 1.

**Graph construction.** For each WSI, we build a graph $G^{(i)} = (X^{(i)}, A^{(i)})$ where the vertices are the tiles features $X^{(i)} \in \mathbb{R}^{n \times d}$, as described in the previous section, and $A^{(i)} \in \mathbb{R}^{n \times n}$ is the adjacency matrix computed based on the euclidean distance between the spatial coordinates of the tiles in the original WSI for a given number $k$ of nearest neighbors.

**Context-Aware Regularization.** To embed the spatial structure of the WSI into the tiles features, we introduce a spatial encoder $f_E$ of parameters $\theta_E$ and a spatial decoder $f_D$

of parameters $\theta_D$ before the classical MIL model $f_{\text{MIL}}$ of (2) as illustrated in Fig. 1

$$f_E \colon \mathbb{R}^{n \times d} \times \mathbb{R}^{n \times n} \to \mathbb{R}^{n \times d_E} \quad f_D \colon \mathbb{R}^{n \times d_E} \to \mathbb{R}^{n \times n} \tag{3}$$
$$(X, A) \mapsto Z, \qquad\qquad Z \mapsto \hat{A}.$$

Here, the goal of $f_E$ is to distill the spatial information contained in an input WSI graph $G = (X, A)$ into tiles features $X$, resulting in a new low-dimensional Context-Aware embedding $Z \in \mathbb{R}^{n \times d_E}$ of the WSI. To achieve this goal, $f_D$ aims at reconstructing the input adjacency matrix $A$ from $Z$ during training while, concurrently, $f_{\text{MIL}}$ aims at predicting the patient risk from $Z$. Formally, this corresponds to adding a Context-Aware Regularization term $\mathcal{L}_{\text{CAR}}$ to the training optimization problem of classical MIL (2):

$$\min_{\theta_{\text{MIL}}, \theta_E, \theta_D} \frac{1}{N} \sum_{i=1}^{N} \left( (1 - \beta)\mathcal{L}_{\text{MIL}}\left( f_{\text{MIL}}(Z^{(i)}; \theta_{\text{MIL}}), y^{(i)} \right) + \beta \mathcal{L}_{\text{CAR}}\left( f_D(Z^{(i)}; \theta_D), A^{(i)} \right) \right) \tag{4}$$

where the same notations as in (2) are used and,

$$\begin{cases} Z^{(i)} = f_E\left(G^{(i)}; \theta_E\right), \\ \mathcal{L}_{\text{CAR}}(\hat{A}, A) = \frac{1}{n^2} \sum_{p,q} \left( A_{pq} \log(\hat{A}_{pq}) + (1 - A_{pq}) \log(1 - \hat{A}_{pq}) \right). \end{cases} \tag{5}$$

By reformulating the training problem as a *joint* optimization task, the models' parameters are obtained by simultaneously solving a pair of objectives, which may initially appear orthogonal. When $\beta = 0$, the method boils down to the classical MIL pipeline, whereas $\beta = 1$ reduces the task to encoding the spatial context only as in a Graph AutoEncoder (GAE) (Kipf and Welling, 2016a). Note that the spatial decoder $f_D$ is not required at inference.

The spatial encoder $f_E$ and spatial decoder $f_D$ are obtained by stacking up, respectively $\ell_E$ and $\ell_D$, graph convolutional network (GCN) layers:

$$\text{GCN}\big((X, A); \theta\big) := A \operatorname{ReLU}(X) W^\theta, \tag{6}$$

where $W^\theta$ is a learnable weight matrix and the matrix $A$ may be preprocessed. If we denote the composition of a function $f$ by itself $t$ times as $f^{\circ t}$, then,

$$Z = f_E(G; \theta_E) := \text{GCN}^{\circ \ell_E}(G; \theta_E), \qquad U_D(Z) = \text{GCN}^{\circ \ell_D}\big((Z, A); \theta_D\big). \tag{7}$$

Similarly to a GAE, the last layer of the spatial decoder is an inner-product decoder with a sigmoid function $\sigma$, that aims at reconstructing the input adjacency matrix $A$,

$$\hat{A} = f_D(Z; \theta_D) := \sigma\big(U_D(Z) U_D(Z)^T\big). \tag{8}$$

## 2.3 DeltaCon for quantitative measure of *Context-Awareness*

Reporting task-related scores alone is not enough to quantify the amount of Context-Awareness in a tiles embedding space. To overcome this issue, we propose a novel *Context-Awareness* metric based on DeltaCon (Koutra et al., 2013).

Let $\hat{Z} \in \mathbb{R}^{n \times p}$ be tiles descriptors, we denote $A(\hat{Z}) \in \mathbb{R}^{n \times n}$ the adjacency matrix based on the $k$ nearest neighbors for the euclidean distance between the components of $\hat{Z}$. This

is different from $A$, defined in sec. 2.2, that is based on the distance between the spatial coordinates of the tiles. $\hat{Z}$ can be any representation of the tiles, here it will either be the features $X$ or the embeddings $Z$ as defined in sec. 2.2.

We propose to use the DELTACON similarity between the adjacency matrix based on tiles descriptors $A(\hat{Z})$ and the adjacency matrix based on the spatial coordinates $A$ for the same WSI as our metric of *Context-Awareness*. Since we have directed graphs and the adjacency matrices $A$ and $A(\hat{Z})$ are non-symmetric, we propose this extension of the original DELTACON similarity

$$\text{DELTACON}\left(A(\hat{Z}), A\right) = \frac{1}{1 + ||S(A(\hat{Z})) - S(A)||_F}, \qquad S(A) := (\mathbb{I} + \epsilon^2 D - \epsilon A)^{-1}, \quad (9)$$

where $D := D_{\text{in}} + D_{\text{out}}$, where $D_{\text{in}}$, resp. $D_{\text{out}}$, is the diagonal matrix counting the incoming, resp. outcoming, edges and $||.||_F$ denotes the Frobenius norm.

Intuitively, $S(A) \in \mathbb{R}^{n \times n}$ is a matrix capturing the degrees of similarity between all tiles when one travels in the original graph corresponding to $A$. Direct neighbors are the most similar tiles, followed by neighbors of neighbors, and so on. As a result, $\text{DELTACON}\left(A(\hat{Z}), A\right)$ is close to 1 when the neighbors in the tiles descriptors space and the spatial coordinates are almost the same and it smoothly decreases towards 0 when those neighbors become dissimilar between the tiles representations and the spatial coordinates.

## 3 Implementation details

### 3.1 Datasets for survival prediction using whole slide images

**Glioblastoma data.** We used H&E slides of patients with glioblastoma from the datasets TCGA GBM (Brennan et al., 2013; McLendon et al., 2008) and TCGA LGG (The Cancer Genome Atlas Research Network, 2015).We filtered cases according to the latest WHO classification for gliomas (Louis et al., 2021). See App. A for more details.

**Colon cancer data.** We used H&E slides of patients with colon adenocarcinoma from TCGA. The TCGA COAD dataset contains a total of 431 cases from 24 centers.

### 3.2 Evaluation

All models were trained and evaluated on TCGA COAD and TCGA GBM using 5-fold nested cross validation (Bengio, 2012) to allow for hyperparameters tuning and assess the generalisation independently on those datasets. Three repeats were used in the inner loop, corresponding to three different random initializations of the MIL or CARMIL models. As a result, metric evaluation on each of the 5 test splits was performed using an ensemble of 15 models (5 inner validation splits and 3 repeats). Ensembling was performed by averaging the risk output of the models of an ensemble. In all the tables, we show the mean (std) C-index for OS obtained using 5-fold nested cross validation. The best results are in bold.

### 3.3 Preprocessing

For tissue segmentation, a 2D U-net (Ronneberger et al., 2015) trained on a pancancer dataset of manually annotated WSIs is used. Tile features of dimension $d = 768$ are

| Model | TCGA COAD | TCGA GBM |
|---|---|---|
| MeanPool | 0.637 (0.02) | 0.621 (0.02) |
| CARMeanPool (ours) | **0.640 (0.05)** | **0.640 (0.03)** |
| ABMIL (Ilse et al., 2018) | 0.635 (0.06) | 0.622 (0.03) |
| CARABMIL (ours) | **0.642 (0.05)** | **0.635 (0.02)** |
| Chowder (Courtiol et al., 2018) | 0.618 (0.07) | **0.642 (0.02)** |
| CARChowder (ours) | **0.666 (0.04)** | 0.634 (0.02) |
| AdditiveMIL (Javed et al., 2022) | 0.645 (0.04) | 0.631 (0.04) |
| CARAdditiveMIL (ours) | **0.648 (0.05)** | **0.638 (0.02)** |
| MultiDeepMIL (Wibawa et al., 2022) | **0.652 (0.05)** | 0.633 (0.03) |
| CARMultiDeepMIL (ours) | 0.649 (0.04) | **0.638 (0.03)** |
| MILTransformer (Shao et al., 2021) | 0.633 (0.05) | **0.639 (0.02)** |
| CARMILTransformer (ours) | **0.664 (0.03)** | 0.629 (0.03) |
| Average MIL models | 0.637 (0.05) | 0.631 (0.03) |
| Average CARMIL models (ours) | **0.652 (0.04)** | **0.636 (0.03)** |

Table 1: Mean C-index (std) on OS performance in MIL models with and without CAR.

computed using a state-of-the-art self-supervised model, Phikon, trained on pancancer H&E slides from TCGA (Filiot et al., 2023). The parameters of Phikon are kept frozen in all our experiments. The preprocessing is the same for all the models we compare to.

### 3.4 Deep learning training

The Cox loss was employed in the supervised training of all models, utilizing overall survival labels. This corresponds to $\mathcal{L}_{\text{MIL}}$ in eq. (4). The CAR loss, denoted as $\mathcal{L}_{\text{CAR}}$ in eq. (5), is applied across all CAR models and we use $\beta = 0.5$ in eq. (4) for the total loss accross all CAR models. The choice of $\beta = 0.5$ followed the empirical observation that $\mathcal{L}_{\text{MIL}}$ and $\mathcal{L}_{\text{CAR}}$ have similar range of values during the first epoch of training. Adam optimizer (Kingma and Ba, 2014) with momentum $\beta_1 = 0.9$ and $\beta_2 = 0.999$ is used for training with a learning rate on the grid $\{0.001, 0.003, 0.01\}$ for all models. The maximum learning rate value 0.01 was chosen heuristically to be the smallest value on the grid for which most MIL models diverged during training. The number of training epochs is optimized on the grid $\{20, 30\}$. One NVIDIA Tesla T4 GPU with 16GB of VRAM and 8 Intel(R) Xeon(R) 2.00GHz CPUs are used for training and inference of each model.

## 4 Experiments

### 4.1 Survival prediction performance

In tables 1 and 2, we report the performance of various MIL models for the challenging task of survival prediction on TCGA COAD and TCGA GBM. In table 1, we compare classical MIL models, selected for being agnostic to the spatial context, to their performance when

| Model | TCGA COAD | TCGA GBM |
|---|---|---|
| MILTransformer with positional encoding | 0.628 (0.07) | 0.636 (0.03) |
| GNN (Li et al., 2018) | 0.635 (0.06) | 0.635 (0.02) |
| LaMIL (Reisenbüchler et al., 2022) | 0.639 (0.04) | **0.640 (0.02)** |
| Average CARMIL models (ours) | **0.652 (0.04)** | 0.636 (0.03) |

Table 2: Mean C-index (std) on OS performance comparison of CARMIL methods to state-of-the art models considering spatial context through their complex architecture.

enhanced with our CAR scheme. In total, CAR improved C-index values by up to 4.8 percentage points (pp) in 9/12 settings and on average the C-index increased by 1.5 pp on TCGA COAD and 0.5 pp on TCGA GBM. In addition, we compare classical MIL models enhanced with CAR to state-of-the-art model architectures that were designed specifically to leverage graphs of tiles. CARMIL models perform similarly or better than these more sophisticated models in terms of C-index; see table 2. This suggests our CAR method improves performance by successfully integrating spatial context through regularization.

### 4.2 *Context-Awareness* performance

In this section, we compile evidence supporting the successful injection of spatial information in the proposed CAR models and its associated performance benefits. First, we assess whether CAR models genuinely exploit the input graph for their predictions. To this end, we perturb the graph at inference time by randomly shuffling all the off-diagonal terms of the adjacency matrix $A$. If the CAR models were to fail to exploit the graph structure, this disruption would not impact their performance. However, the results reported in table 3 indicate that graph shuffling leads to a degradation in the model's performance, showcasing that the graph structure is indeed used in the model's decision-making process.

Secondly, we assess the *Context-Awareness* of trained CARMIL models compared to the feature extractor, using the DELTACON metric that we defined in sec. 2.3. For each WSI, the adjacency matrix computed in the embedding space learnt by the spatial encoder of fig. 1 is compared to the original adjacency matrix that accounts for the spatial arrangement of the tiles within the slide. The DELTACON provides us with a score between 0 and 1. A higher DELTACON score indicates a higher degree of spatial consistency in the learnt embedding space as it implies that the arrangement in the embedding space closely aligns with the original spatial organization within the slide. We average this slide-level score across the whole TCGA COAD and GBM datasets, see table 4. In app. B.4, we showcase an example of a WSI illustrating how the embeddings provided by CARMIL encoders are more spatially consistent than the original features. Moreover, in fig. 2, we can clearly see how the spatial encoders implement more *Context-Awareness* than the feature extractor (dotted lines). Indeed, DELTACON is almost always greater for CARMIL models than for the feature extractor. Therefore, these findings confirm the successful injection of spatial knowledge resulting from the CAR. Additionally, we explored the relationship between spatial information incorporated by the spatial encoder and C-index performance. In TCGA

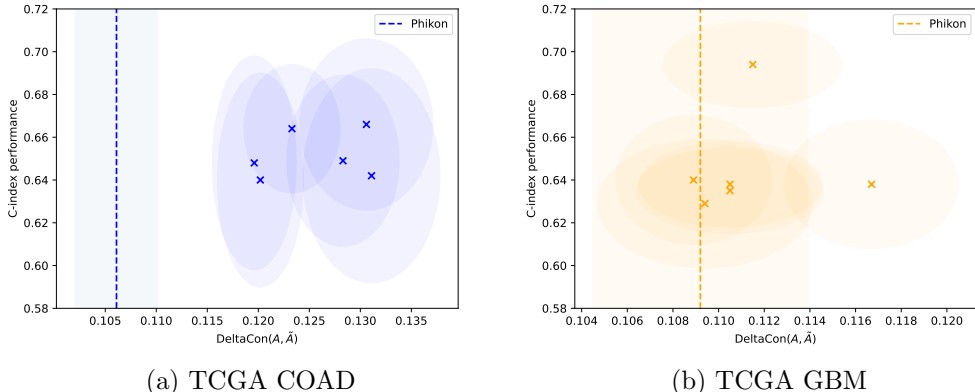

(a) TCGA COAD        (b) TCGA GBM

Figure 2: Spatial information retained in CARMIL models with C-index performance. *Context-Awareness* is quantified using DeltaCon similarity between $A$ and $A(Z)$; see sec. 2.3. Each point represents an ensemble of CARMIL models from nested cross-validation, with filled areas indicating one standard deviation around the mean. The vertical dotted line shows the average *Context-Awareness* in the original features.

COAD, the performance seems to correlate with the level of *Context-Awareness*, potentially providing insights into colon cancer. However, in TCGA GBM, it is uncertain whether adding spatial elements improves performance, and understanding how *Context-Awareness* influences overall performance remains an open question.

## 5 Conclusion

The MIL framework fails to leverage spatial interaction and organization of tiles in WSIs, potentially limiting prognostic model performance. In this work, we proposed to tackle this issue by injecting spatial knowledge into the traditional MIL framework exclusively through the prism of regularization. We introduced CARMIL, a method to embed spatial relations between tiles directly into tile features. This addition mimics the pathologist's consideration of spatial arrangement in slide-level prognosis. We evaluated our method on survival prediction for colon cancer and glioblastoma, showing improved performance and revealing performance declines when spatial context is disregarded. We also introduced a metric for quantifying *Context-Awareness*, hoping to help researchers assess spatial consistency more systematically. Lastly, we discussed in App. C the influence of certain parameters on our method, but we did not investigate the impact of the feature extractor on overall performance. The relationship between model robustness and *Context-Awareness* remains an open question, suggesting avenues for future research.

## Acknowledgments and Disclosure of Funding

We thank Jean Philippe Vert and Eric Durand for their thorough proofreading of our paper, as well as Jean-Baptiste Schiratti and Alexandre Filiot for providing insightful feedback on various aspects of this work. We are thankful to Céline Thiriez too for her encouragement and support.

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

| | TCGA COAD | | TCGA GBM | |
|---|---|---|---|---|
| Model | Original | Shuffled | Original | Shuffled |
| CARMeanPool | 0.640 (0.05) | **0.641 (0.04)** | **0.640 (0.03)** | 0.631 (0.02) |
| CARABMIL | 0.642 (0.05) | **0.643 (0.05)** | **0.635 (0.02)** | 0.629 (0.01) |
| CARChowder | **0.666 (0.04)** | 0.661 (0.04) | **0.634 (0.02)** | 0.613 (0.01) |
| CARMultiDeepMIL | 0.649 (0.04) | **0.651 (0.05)** | **0.638 (0.03)** | 0.630 (0.02) |
| CARAdditiveMIL | 0.648 (0.05) | **0.656 (0.05)** | **0.638 (0.02)** | 0.633 (0.02) |
| CARMILTransformer | **0.664 (0.03)** | 0.633 (0.04) | **0.629 (0.03)** | 0.607 (0.02) |
| Average CARMIL | **0.652 (0.04)** | 0.648 (0.05) | **0.636 (0.03)** | 0.624 (0.02) |

Table 3: Ablation of CARMIL models performance with and without shuffling the input graphs during inference after training with unperturbed graphs. This measures the contribution of the spatial *Context-Awareness* to the C-index.

## Appendix A. Glioblastoma WHO 2021 classification

The WHO 2021 classification defines glioblastoma as IDH-wild type and H3-wild type brain tumor with at least one of the following features: necrosis and/or microvascular proliferation, TERT promoter mutation, EGFR amplification, or concomitant gain of chromosome 7 and loss of chromosome 10. We will refer as TCGA GBM to those glioblastoma cases from the datasets TCGA GBM and TCGA LGG after reclassification. TCGA GBM contains 352 cases from 18 centers.This change of classification of glioblastoma has been shown to have a negative impact on the prognostic value of previously published biomarkers (Zakharova et al., 2022). Therefore, it is clinically important to evaluate previous and new prognostic models on glioblastoma using the new WHO classification.

## Appendix B. Reproducibility

### B.1 Construction of the spatial adjacency matrix $A$

Each WSI $G = (X, A)$ consists of a set of $n$ tiles. For each tile $p$, we keep track of its spatial coordinates in the 2D plane formed by the tissue region, $\mathbf{c}_p = (x_p, y_p)$, in addition to the $d$-dimensional features vector $\mathbf{u}_p$ produced by the feature extractor for that tile. We thus compute the gaussian kernel, $\forall p \neq q$,

$$K_{pq} = \exp\left(-\frac{||\mathbf{c}_p - \mathbf{c}_q||_2}{2}\right),$$

and we set the diagonal to 0. Provided $k$ the number of neighbors, we select the $k$ nearest neighbors for each node based on the similarity matrix $K$. The adjacency matrix $A$ is defined such that $A_{pq}$ equals $K_{pq}$ if tile $q$ is one of the nearest neighbors of tile $p$, and 0 otherwise.

## B.2 Construction of the adjacency matrix $\tilde{A}$ in the embedding space

Adjacency matrices in the embedding space are used for the evaluation of *Context-Awareness* using DELTACON.

Consider training of our model complete and the final parameters to have converged to $\theta_{\mathrm{MIL}}, \theta_E, \theta_D$. Given a WSI, represented as $G = (X, A)$, our spatial encoder returns a lower dimensional vector $Z = f_E(G; \theta_E) \in \mathbb{R}^{n \times d_E}$. Observe that $Z_p$ is the $d_E$-dimensional embedding for tile $p$. Based on this vector, we construct an adjacency matrix $\tilde{A}$, following the same principle as before, but this time based on the affinity between embeddings rather than using the spatial coordinates of the tiles, namely, $\forall p \neq q$,

$$\tilde{K}_{pq} = \exp\left(-\frac{||Z_p - Z_q||_2}{2}\right),$$

and we set the diagonal to 0. We similarly pick the $k$ nearest neighbors for each tile and we construct the adjacency matrix $\tilde{A}$, that we refer to as the adjacency matrix in the embedding space. Therefore, $\tilde{A}_{pq}$ is the affinity between the embeddings of the tiles $p$ and $q$.

## B.3 Computation of DeltaCon

Observe that both adjacency matrices, $A$ and $\tilde{A}$, are non-symmetric. Indeed, they are derived by taking the $k$-nearest neighbors of each node in the kernel matrices $K$ and $\tilde{K}$, which is an inherently non-symmetric operation. In simple words, tile $i$ can be connected to tile $j$, but $i$ is not necessarily a neighbor of $j$, meaning the induced graphs are directed. In Koutra et al. (2013), the authors introduce a proxy function defined for undirected graphs with symmetric adjacency matrix $A$ and degree matrix $D$ as

$$S(A) \coloneqq (\mathbb{I} + \epsilon^2 D - \epsilon A)^{-1}. \tag{10}$$

We extend this definition to directed graphs by considering a non-symmetric adjacency matrix $A$ and a degree matrix that accounts for the number of edges entering or leaving each node. Namely, considering $D \coloneqq D_{\mathrm{in}} + D_{\mathrm{out}}$, where $D_{\mathrm{in}}$, resp. $D_{\mathrm{out}}$, is the diagonal matrix counting the incoming, resp. outcoming, edges, then we can refer back to eq. (10) to generalize the definition of DELTACON similarity between directed graphs. Given $A$ and $\tilde{A}$, as described in sec. B.1 and B.2, and their associated degree matrices $D$ and $\tilde{D}$, we can now quantitatively assess the amount of spatial information, or *Context-Awareness*, of any CARMIL model by computing,

$$\text{DELTACON}(A, \tilde{A}) \coloneqq \frac{1}{1 + ||S(A) - S(\tilde{A})||_F}. \tag{11}$$

In table 4, we assess the *Context-Awareness* of the best-performing CARMIL model from the 15 models evaluated through nested cross-validation, see sec. 3.2. For each WSI in each dataset, we first compute the spatial adjacency matrix $A$ with $k = 8$ nearest neighbors, as in sec. B.1. Then, for all rows in table 4 except the first, the matrix $\tilde{A}$ is computed based on the embedding vector $Z$ from the model's encoder after training, also with $k = 8$, see sec. B.2. For the first row, the adjacency matrix $\tilde{A}$ is based on the feature vector $X$

Table 4: Correlation between the DeltaCon similarity that quantifies the *Context-Awareness* of MIL models and their associated C-index performance.

| Model | TCGA COAD | | TCGA GBM | |
|---|---|---|---|---|
| | DeltaCon | C-index | DeltaCon | C-index |
| Feature extractor | 0.1061 (0.0041) | | 0.1092 (0.0047) | |
| CARMeanPool | 0.1202 (0.0042) | 0.640 (0.05) | 0.1089 (0.0034) | 0.640 (0.03) |
| CARABMIL | 0.1311 (0.0067) | 0.642 (0.05) | 0.1105 (0.0041) | 0.635 (0.02) |
| CARChowder | 0.1306 (0.0065) | 0.666 (0.04) | 0.1115 (0.0039) | 0.634 (0.02) |
| CARAdditiveMIL | 0.1196 (0.0041) | 0.648 (0.05) | 0.1105 (0.0040) | 0.638 (0.02) |
| CARMultiDeepMIL | 0.1283 (0.0055) | 0.649 (0.04) | 0.1117 (0.0038) | 0.638 (0.03) |
| CARMILTransformer | 0.1233 (0.0047) | 0.664 (0.03) | 0.1094 (0.0047) | 0.629 (0.03) |

generated by the feature extractor. The similarity score given by eq. (11) is averaged across all slides within each dataset to produce the table's values, along with the C-index mean performance of the corresponding model. In fig. 2, we report these same results in a scatter plot to better illustrate two important findings. First, by comparing the scores from the feature extractor to the other models on both datasets, we confirm the successful injection of *Context-Awareness* into our CARMIL models. As a side note, it is worth mentionning here that, even though DeltaCon lies between 0 (indicating very dissimilar graphs) to 1 (for identical graphs), the interpolation between these extremes does not seem to evenly spread out within the interval $[0, 1]$. This would explain the small variations in our table 4. In fact, as evidenced in the tables of results from the original paper Koutra et al. (2013), most values are closer to 0 than 1. Nevertheless, for our analysis, only the relative ordering of these quantities is key in proving the enhanced spatial awareness provided by our method. Second, we try to correlate gains in performance with the amount of added *Context-Awareness* in the CARMIL model. Conclusions such as the more spatial knowledge is incorporated, the better the model perform, are rather difficult to draw.

## B.4 Qualitative assessment of *Context-Awareness* in CARMIL models

Consider a WSI, from which $n$ tiles are sampled. Passing this slide to our feature extractor, we get a vector $X \in \mathbb{R}^{n \times d}$. For each tile, we compute the mean of the eucliden distance between the $d$-dimensional features of this very tile with the features of all its $k = 8$ spatial nearest neighbors within the slide. This results in a vector $Y^{\text{features}} \in \mathbb{R}^n$ that accounts for how well aligned the features learnt by the feature extractor conform to the spatial arrangement of the slide. If the features exactly reflect the spatial context of each tile, the coefficients in the vector $Y^{\text{features}}$ should be fairly constant and of low magnitude. Their variations are shown in fig. 3a, where we superposed the underlying slide with the values of $Y^{\text{features}}$. Next, we pass these features through our CARMIL encoder at inference time, following the same procedure but using the embedding vector $Z \in \mathbb{R}^{n \times d_E}$ – with the same notation as before. This results in a vector $Y^{\text{CARMIL}} \in \mathbb{R}^n$. We report the coefficients in

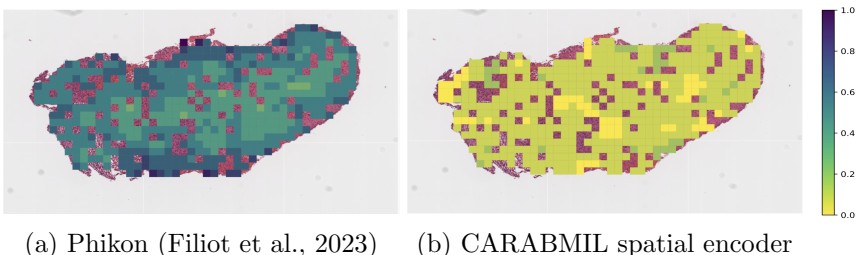

(a) Phikon (Filiot et al., 2023)    (b) CARABMIL spatial encoder

Figure 3: Heatmap of mean euclidean distances between the representation of each tile given by (a) Phikon, the feature extractor, and (b) CARABMIL spatial encoder to its 8 spatial nearest neighbors. All mean distances were scaled together to $[0, 1]$. The WSI is taken from TCGA GBM.

fig. 3b. To obtain similarly scaled heatmaps, we jointly normalize the coefficients of both matrices by their maximum value, ensuring all their coefficients range between 0 and 1. Note that both tile features $X$ and tile embeddings $Z$ are first normalized to have magnitude 1, ensuring a fair comparison that mitigates high-dimensional concentration effects on vector norms.

From fig. 3, we can clearly see that the embeddings generated by the CARMIL encoder evolve smoothly in the 2d plane, resulting in a uniformly low heatmap value across the slide. In comparison, the tile features provided by the feature extractor Phikon exhibit less alignment with the spatial organization of the tiles, providing another qualitative indication of the enhanced spatial consideration in CARMIL models.

## Appendix C. Discussion on certain parameters

**Number of tiles $n$.** The number of tiles $n$ that we randomly select from the tissue region of each WSI has a significant impact, and we provide an intuitive explanation for this. The amount of spatial context required to make accurate predictions is inherently tied to the number of tiles, as it determines the scale at which patterns can be grouped. Moreover, we observe that different WSIs can vary greatly in size, resulting in a wide range of number of tiles across datasets. Whilst we fixed $n$ regardless of the WSI at stake, it would be a natural improvement to choose $n$ for each slide in proportion to the original size of the tissue region.

**Number of nearest neighbors $k$.** Similar to the number of tiles, it is probably advisable to choose $k$, the number of nearest neighbors, in proportion to the number of tiles $n$, and subsequently, the size of the WSI. In our experiments, we optimized $k$ across a fixed grid of values, independent of $n$.

**Number of GCN layers $\ell_E, \ell_D$.** The number of layers $\ell_E$ and $\ell_D$ in the spatial encoder and decoder, as well as the dimensions for the projections they induce were part of our grid search. We observed that $(\ell_E, \ell_D) = (1, 1)$ worked best for TCGA COAD and $(\ell_E, \ell_D) = (2, 2)$ for TCGA GBM. The dimensions of each layer were also finetuned from a grid of fixed values, resulting in $d_E = d$ as an optimal choice for the spatial encoder and all intermediate

layers. Besides, we observed that significantly reducing the embedding dimension $d_E$ almost systematically led to a substantial decline in performance.

