# OpenReview forum: "CARMIL: Context-Aware Regularization on Multiple Instance Learning models for Whole Slide Images"
_MICCAI.org/2024/Workshop/COMPAYL — COMPAYL 2024_

### Official Review · Reviewer_XxFx · 2024-07-09
**Review of paper 15: A new method for incorporating spatial information into classic MIL**

**Custom Rating:** 4
**Confidence:** 4

**Review:**

The authors proposed a new method for considering the spatial information between patches (instances) in the classic multiple instance learning (MIL) approach.
They incorporated encoder-decoder modules along with graphs to classic MIL. The work is novel and original, and the paper is well-written. They explained the method and its novelty appropriately, making it clear except for the model formulation, where the mathematics are complex and hard to follow. It would be beneficial to include a table of annotations explaining the symbols and their types separately. The results show that the proposed method can perform better than classic MIL, indicating the quality of the work. However, there is no information on the amount of data used for training, validation, and testing. The authors must declare this information. The significance of the work can be further discussed, since the results showed only a slight improvement, especially given the lack of information about the number of data used for evaluation and the data selection process.

Pros:
- The manuscript is clear and well-written.
- The proposed method is easy to understand.

Cons:
- There is no information about the number of data used for training, validation, and testing.
- The comparison results show only a slight improvement over other available MIL methods, both with and without CAR, and the state-of-the-art method for one dataset.

There are some minor comments:
- The term "multiple instance learning (MIL)" is abbreviated twice in the text.
- The authors should explain the tables clearly, indicating, for instance, that the results are averages of five-fold cross-validation and that if the parentheses show standard deviations or bold numbers show the best results.

---

### Official Review · Reviewer_nfD7 · 2024-07-10
**Review of CARMIL: Context-Aware Regularization on Multiple Instance Learning models for Whole Slide Images**

**Custom Rating:** 3
**Confidence:** 4

**Review:**

The authors propose a method that leverages spatial information for multiple-instance-learning (MIL) in whole slide images (WSI) with graph neural networks (GNN). The method can be applied to existing MIL frameworks. The method uses a GNN encoder to contextualize patch embeddings, and a decoder to reconstruct the adjacency matrix. A metric to evaluate context awareness is also proposed. The method is evaluated for survival analysis in two datasets.

Pros:
- The authors use GNNs and a reconstruction loss to contextualize patch embeddings to address MIL tasks.
- They also propose a metric to analyze the spatial awareness of the embeddings.
- Their method is applicable to existing MIL frameworks and they show better performance.
- The paper is well-written and easy to follow.

Cons/comments:
- Using a GNN encoder to contextualize patch representations is not new. Whereas the authors claim that the novelty of their method lies in adding the regularization (reconstruction with an autoencoder), they don't analyze whether the reconstruction term actually boosts the performance or not. Couldn't it be that the boost in performance is solely due to the GNN encoder?
- What patch feature extractor is being used? With the rise of foundational models for digital pathology, the evaluation of some of them should be explored.

Minor comments:
- Eq 6 (GCN) misses the normalization of the adjacency matrix

---

### Official Review · Reviewer_RDcy · 2024-07-12
**interesting context encoding for MIL, but results could be improved**

**Custom Rating:** 4
**Confidence:** 3

**Review:**

**Overview**
In this paper, the authors challenge the assumption in Multiple Instance Learning (MIL) that instances are independent, and propose to include spatial awareness into MIL through regularization. This is done through Graph Neural Networks and an auxiliary loss function. The motivation is biologically sound and the proposed method contains novelty. However, the method seems highly dependent on sampling, implying that some data needs to be discarded, even at inference time, which can be sub-optimal.

**Pros**
In general, the text is well-written and easy to follow.
The Context-aware regularization module based on Graph AutoEncoders contains novelty.
Results in section 4.2 are interesting in the sense that authors try to show that the proposed method works as intended and integrates spatial information.

**Cons**
The graph is constructed over sampled patches and the context-aware loss works by predicting the adjacency matrix. This seems to work because of sampling and discarding part of the data. Otherwise, if the authors use all patches, the auxiliary task may become too trivial to be useful. This implies that 1) inference may change across runs due to sampling, and 2) key regions may be discarded.
In the results (table 1), improvements in the proposed method are generally small.


**Further comments**

Data:
The data size of Glioblastoma should be in the main text. Please, include this information.

Results:
In Table 1, the improvements of the proposed method are generally small compared with the corresponding baseline, especially within COAD. The largest improvement is when compared to Chowder in COAD, but it is worse in GBM.
Similarly, in Table 2, results do not show a clear benefit of the proposed method over other aggregations of spatial context, especially in GBM.
In Figure 2, it is nice to see the authors trying to validate the assumptions of the model about encoding spatial context. The reviewer also appreciates that the authors detail the DeltaCon similarity is not linear. Still, the values between the different models are small in magnitude. So, the reviewer wonders if DeltaCon really is informative.

---

### Decision · Program_Chairs · 2024-07-16

Accept